# Isolation, Characterization, and Autophagy Function of BECN1-Splicing Isoforms in Cancer Cells

**DOI:** 10.3390/biom12081069

**Published:** 2022-08-02

**Authors:** Chinmay Maheshwari, Chiara Vidoni, Rossella Titone, Andrea Castiglioni, Claudia Lora, Carlo Follo, Ciro Isidoro

**Affiliations:** Laboratory of Molecular Pathology, Department of Health Sciences, Università del Piemonte Orientale “A. Avogadro”, 13100 Novara, Italy; chinmay.maheshwari@uniupo.it (C.M.); chiara.vidoni@med.uniupo.it (C.V.); rossellatitone@hotmail.com (R.T.); andrea.castiglioni@ieo.it (A.C.); clora@studenti.uninsubria.it (C.L.)

**Keywords:** Beclin 1, alternative splicing, autophagy, mitophagy, Bcl-2, isoforms, BH3 domain breast cancers, ovarian cancers

## Abstract

Alternative splicing allows the synthesis of different protein variants starting from a single gene. Human Beclin 1 (*BECN1*) is a key autophagy regulator that acts as haploinsufficient tumor suppressor since its decreased expression correlates with tumorigenesis and poor prognosis in cancer patients. Recent studies show that BECN1 mRNA undergoes alternative splicing. Here, we report on the isolation and molecular and functional characterization of three *BECN1* transcript variants (named BECN1-α, -β and -γ) in human cancer cells. In ovarian cancer NIHOVCAR3, these splicing variants were found along with the canonical *wild-type*. BECN1-α lacks 143 nucleotides at its C-terminus and corresponds to a variant previously described. BECN1-β and -γ lack the BCL2 homology 3 domain and other regions at their C-termini. Following overexpression in breast cancer cells MDA-MB231, we found that BECN1-α stimulates autophagy. Specifically, BECN1-α binds to Parkin and stimulates mitophagy. On the contrary, BECN1-β reduces autophagy with a dominant negative effect over the endogenous *wild-type* isoform. BECN1-γ maintains its ability to interact with the vacuolar protein sorting 34 and only has a slight effect on autophagy. It is possible that cancer cells utilize the alternative splicing of BECN1 for modulating autophagy and mitophagy in response to environmental stresses.

## 1. Introduction

Human Beclin 1 *(BECN1)* is a haploinsufficient tumor suppressor gene located on chromosome 17q21 breast cancer tumor susceptibility locus in close proximity to *BRCA1* [1,2]. *BECN1* is found mono-allelically deleted in about 40% to 75% of human sporadic breast cancers and ovarian cancers. In vivo studies in mice indicate that loss of heterozygosity for *BECN1* leads to an increased incidence of spontaneous carcinomas [3,4], including breast carcinomas with basal-like features [5]. Clinical data further confirm that low BECN1 protein levels are associated with poor prognosis and rapid progression of breast cancer [6]. Moreover, data from patients with ovarian cancer and lymphomas indicate that higher levels of BECN1 are associated with a better prognosis and higher survival rate [7,8]. Interestingly, patients with ovarian cancer presenting co-deletions of *BECN1* and *BRCA1* were found to benefit from platinum-based therapy and show improved chances of survival [9]. Overall, these studies highlight the importance of BECN1 as both tumor prognostic marker and potential molecular target for the therapeutic treatment of cancer.

Originally isolated as an interactor of the anti-apoptotic BCL2 [1], BECN1 is a key signaling hub for both apoptosis and autophagy. BECN1 collects signals and balances apoptosis with autophagy through its multi-domain protein structure comprising a BCL2-homology 3 domain (BH3D) [10], a coiled-coil domain (CCD) [11], and an autophagy-specific BARA region [12] that includes an evolutionarily conserved domain (ECD) [13]. The role of BECN1 in the regulation of autophagy has been extensively investigated [14]. In this study, we focus on understanding the role of different BECN1 splicing isoforms on autophagy. Autophagy is an evolutionarily conserved catabolic process acquired by the eukaryotic cell for the elimination/recycling of potentially hazardous cytosolic components (*cargo*) through lysosomal degradation. Together with the phosphatidylinositol-3 kinase Class III vacuolar protein sorting 34 (PI3KCIII/VPS34), the autophagy-related homolog 14 (ATG14), and the vacuolar protein 15 (VPS15), BECN1 forms the PI3KCIII complex-I that initiates the nucleation of the phagophore [15]. Following the substitution of ATG14 with the UV-radiation resistance-associated gene (UVRAG), the BECN1/PI3KCIII complex-II promotes both autophagosome maturation and endosomal trafficking [16,17,18]. Studies in neurodegenerative disorders have also revealed an interaction between BECN1 and the mitochondrial membrane proteins PTEN-induced kinase 1 (PINK1) and E3-ubiquitin ligases Parkin (PRKN) that promotes the selective sequestration of compromised mitochondria, or mitophagy [19,20]. In cancer, the role of autophagy remains controversial, with autophagy reported to have a putative dual role, acting either as a tumor promoter or suppressor depending on the genetic and epigenetic signature of the cancer cells and the tumor microenvironment context [21]. Among the proteins regulating autophagy, the transcription factor EB (TFEB) plays a key role in regulating the expression of both autophagy and lysosomal genes [22]. By stimulating lysosomal biogenesis, TFEB may play either a tumor-promoter role or -suppressor role. In fact, high lysosomal biogenesis can either facilitate the induction of apoptosis by the release of cathepsins from the lysosomal compartment [23,24], or support a high level of autophagy for the degradation of damaged organelles and the productions of cellular component necessary to sustain tumor growth [25]. Considering its role in balancing apoptosis with autophagy, BECN1 may play a dual role in cancer too by either stimulating apoptosis or autophagy [26,27].

In the human genome, almost 95% of early pre-mRNA transcripts undergo the post-transcriptional modification known as alternative splicing (AS). AS refers to the combinatorial rearrangement and/or confiscation of exons and introns (or parts of them) generating a multitude of transcripts from a small pool of human genes, hence escalating the proteomic diversity. This molecular event is responsible for the production of different protein isoforms that may even work antagonistically [28,29]. Transcriptomic studies in the field of neurodegenerative disorders such as Parkinson’s and Alzheimer’s diseases show that anomalous defects of the splicing machinery aggravate the underlying conditions [30]. AS plays a role in shaping the tumor phenotype too, as shown by a differential exon usage in tumor cells compared to normal cells [31]. Recent pan-cancer studies exploiting patient databases such as The Cancer Genome Atlas (TCGA) reveal that the presence of aberrant splicing machinery genes correlates with breast cancer aggressiveness [32,33]. AS is known to alter autophagy too by generating alternative splicing isoforms of autophagy-related proteins including ATG14, WIPI1, WIPI2, ATG10, ATG12, ATG16L1, ATG16L2, ATG7 and the ATG8 gene family [34]. In addition to these autophagy-related proteins, the BECN1 pre-mRNA undergoes AS as well. To date, two different short BECN1-splicing variants have been reported, respectively, in leukemia cells and HeLa cells [34,35,36].

Our study extends the current knowledge on the splicing of BECN1 by unveiling two novel mRNA splicing variants of BECN1. These novel BECN1 isoforms (here referred to as BECN1-β and -γ) were isolated from the ovarian cancer cell line. NIHOVCAR3. BECN1-β and -γ were found to be co-expressed in the same donor cell line along with the canonical *wild-type* isoform (BECN1-*wt*) and with a fourth isoform (here referred to as BECN1-α) already found by another group in leukemia cells [35]. Following exogenous expression in a different cancer cell line, isoforms α, β and γ were characterized by subcellular localization and immunoprecipitation studies, and their effect on autophagy was assessed by analysis of LC3 and p62 turnover assay. Our results show that different BECN1 isoforms can have diverse effects on autophagy, underlining the importance to investigate the presence of alternative splicing isoforms of BECN1 in cancer and their role in autophagy.

## 2. Materials and Methods

### 2.1. Cell Cultures

Ovarian cancer cells NIHOVCAR3 (HTB-161, ATCC, Manassas, VA, USA) and A2780 (93112519, ECACC, Salisbury, UK), and triple-negative metastatic breast cancer cells MDA-MB231 (HTB-26, ATCC) were cultured, respectively, in RPMI-1640 (R8758, Sigma-Aldrich, St. Louis, MO, USA) and DMEM (D5671, Sigma-Aldrich). The culture media were supplemented with 10% FBS (ECS0180L, South America origin, EuroClone, Milano, Italy), 1% *v*/*v* penicillin-streptomycin (P0781, Sigma-Aldrich), and 1% *v*/*v* glutamine (G7513, Sigma-Aldrich). All the cell lines were grown under standard conditions at 37 °C with 5% CO_2_ and employed in our experiments between passages 3 and 10.

### 2.2. Cloning of BECN1 mRNA Isoforms

A2780 and NIHOVCAR3 cells were plated as monolayers at 5 × 10^4^ cell/cm^2^ and total poly A^+^ mRNA was purified with the Oligotex Direct mRNA extraction kit (72022, Qiagen, Hilden, Germany) following the manufacturer’s instructions. Cloning of BECN1 isoforms, as described below, was repeated also starting from the total RNA purified with the TRIzol reagent (T9424, Sigma-Aldrich) from the same cell lines and confirmed the isoforms isolated starting from the total poly A^+^ mRNA (not shown).

Total cDNA was reversely transcribed from the purified total mRNA/RNA using the RevertAid First Strand cDNA synthesis Kit (K1622, Thermo-Scientific, Waltham, MA, USA). BECN1 cDNA was then amplified by PCR using the *Taq* DNA polymerase recombinant (10342-020, Invitrogen, Waltham, MA, USA) and primers designed to anneal at the 5′- and 3′-UTR of BECN1 (Appendix A, BECN1 cloning primers). These primers were designed for annealing the sequence of the canonical *wild-type* BECN1 transcript variant of 2098 nt (GenBank: AF139131) [2]. BECN1 primers included also additional nucleotides at their 5′-termini for either EcoRI (F-hBECN1) or BamHI (R-hBECN1) restriction enzyme digestion required for the directional cloning into the vectors described below.

The PCR product (expected ~1981bp for the *wild-type* transcript) was then purified by agarose gel electrophoresis and gel extraction using the QIAEX II gel extraction kit (20021, Qiagen). The gel-purified PCR product was then digested with the restriction enzymes EcoRI and BamHI and subcloned into the pcDNA 3.1/Zeo (−) vector (Invitrogen) vector using T4 DNA ligase (15224-017, Invitrogen) (Appendix A). After bacterial transformation, the colonies were screened for the presence of BECN1 cDNA by PCR. The screening PCR was performed using the *Taq* DNA polymerase recombinant and primers designed for annealing the sequences downstream from the previous primers set (Appendix A, BECN1 internal primers F-908 and R-1674). The identified positive clones were also checked by restriction enzyme digestion with BglII to confirm that BECN1 cDNA was inserted in the plasmid in the correct orientation (Appendix A).

To confirm the specific amplification of BECN1 cDNA, we also performed a nested PCR using the gel-purified PCR product as a template. Nested PCR was performed using the DyNAzyme EXT DNA polymerase (F505L, Fisher Scientific, Waltham, MA, USA) and the BECN1 internal primers F-908 and R-1674. The nested PCR product was analyzed by agarose gel electrophoresis.

Positive clones were subjected to DNA sequencing analysis by Sanger direct sequencing method employing BigDye Terminator v1.1 Cycle Sequencing kit (Applied Biosystems, Waltham, MA, USA) following the manufacturer’s instructions. The primers employed for the cycle sequencing are shown in Appendix A (BECN1 internal primers). The products of cycle sequencing PCR were purified by Ethanol/EDTA/Sodium Acetate precipitation and subjected to automatic sequencing (ABI PRISM 3100, Applied Biosystems). Sequence analysis confirmed the correct insertion of BECN1 cDNA into the pcDNA 3.1/Zeo (−) vector and allow us to identify 4 different BECN1 isoforms (BECN1-α, -β, -γ and *wild-type*). Of note, the 4 isoforms were isolated exclusively from NIHOVCAR3. Only the canonical *wild-type* isoform of BECN1 together with the known variant BECN1-α were isolated from A2780 (not shown).

The identified BECN1 isoforms were subcloned also into the pEGFP-N1 vector (Clontech, Mountain View, CA, USA) following digestion with EcoRI and BamHI restriction enzymes (Appendix A). The resulting vectors codify for each BECN1 isoforms fused with the GFP and were employed for the immunofluorescence and immunoprecipitation studies. Sequencing analysis was performed as described above to confirm the in-frame insertion of each BECN1 cDNA at the N-terminus of the GFP cDNA.

### 2.3. Transient Transfections and Treatments

BECN1 isoforms fused with the GFP were transiently expressed in MDA-MB231 cells. Transient transfections were performed with Lipofectamine P-3000 (L3000-15, Invitrogen) following the manufacturer’s instructions. Briefly, confluent cells were incubated for 6 h with Lipofectamine-P3000 solutions containing 6 µg of plasmid and were cultured for 36 h to allow the expression of the exogenous BECN1. When indicated, the cells were treated with 30 µM Chloroquine or Clq (C6628, Sigma-Aldrich) for 4 h for the LC3 or p62 turnover assay, or with 10 µM of the uncoupler cyanide m-chlorophenylhydrazone or CCCP (C2579, Sigma-Aldrich) for 3 h to induce mitophagy.

### 2.4. Western Blotting Analysis

Cell monolayers were washed with PBS 1x and harvested in RIPA buffer (0.5% Sodium Deoxycholate, 1% NP-40, 0.1% Sodium Dodecyl Sulfate in PBS solution) containing protease inhibitor cocktail and phosphatase inhibitors (Na_3_VO_4_ and NaF). Cells were homogenized using a Microson^TM^ XL 2000 ultrasonic cell disruptor (Misonix Inc., Farmingdale, NY, USA) and protein concentration was determined with the NanoDrop^TM^ 2000 (Thermo-Scientific). Equal amounts of protein homogenates (30 μg) were denatured in Laemmli buffer, separated by SDS-PAGE electrophoresis and transferred onto a PVDF membrane (Bio-Rad, Hercules, CA, USA). Membranes were blocked with 5% nonfat dry milk for 1 h at room temperature and probed overnight at 4 °C with the indicated primary antibodies. Membranes were then probed with peroxidase-labeled secondary antibodies. Chemiluminescent signal was detected by luminol solution (PerkinElmer Inc., Waltham, MA, USA) and visualized using the ChemiDoc XRS Imaging System (Bio-Rad). Densitometric analysis was performed to assess the intensity of the bands using the Quantity One Software (Bio-Rad).

### 2.5. Immunofluorescence

For immunofluorescence studies, cells were plated at 4 × 10^4^ cells/cm^2^ on coverslips, transiently transfected with the vectors expressing the BECN1 isoforms, and allowed to reach confluency. Coverslips were then washed with PBS, fixed with ice-cold methanol, and permeabilized with 0.2% Triton X-100 in PBS. Subsequently, cells were incubated with the primary antibodies overnight at 4 °C in a humid chamber. The following day, the coverslips were washed with 0.1% Triton X-100 in PBS and incubated for 1 h at room temperature with the appropriate secondary antibodies. Nuclei were stained with DAPI. Following 3× washes with 0.1% Triton X-100 in PBS, the coverslips were mounted onto glass slides using Slow-FADE antifade reagent (Life Technologies, Carlsbad, CA, USA) and fluorescence images were acquired at an objective magnification strength of 63X with a Leica DMI6000 fluorescence microscope (Leica Microsystems AG, Wetzlar, DE). BECN1 isoforms were imaged with an anti-GFP antibody. Fluorescence signal intensities were quantified with the ImageJ 1.48v (NIH) software.

For the Mitotracker assay, cells were plated on coverslips at 4.0 × 10^4^ cells/cm^2^ and transfected with the pEGFP-N1 plasmids carrying BECN1-wt or -α. Cells were then incubated with 500 nM Mitotracker™ RED FM (M22425, Invitrogen) for the last 15 min at 37 °C. Coverslips were then washed with PBS, fixed with 4% paraformaldehyde and permeabilized with 0.2% Triton X-100 in PBS. Subsequently, cells were incubated with the LC3 antibody and processed for immunofluorescence staining as described above.

### 2.6. Co-Immunoprecipitation

Cells were seeded in 60 mm culture dishes at 4.5 × 10^4^ cells/cm^2^ and transfected with the pEGFP-N1 plasmids carrying the BECN1 isoforms. Before harvesting the cells, each culture dish was treated with the cross-linker dithiobis succinimidyl propionate (DSP, D3669, Sigma Aldrich) for 15 min at 37 °C. Cells were then harvested in RIPA buffer (0.5% Sodium Deoxycholate, 1% NP-40, 0.1% Sodium Dodecyl Sulfate in PBS solution) containing a protease inhibitor cocktail and phosphate inhibitors (Na_3_VO_4_ and NaF). Equal amounts of protein homogenates (500 µg) were incubated overnight at 4 °C with an anti-GFP antibody (5 μL). The antibody–antigen complexes were then precipitated using the Protein G Sepharose^®^ 4 Fast Flow (17-0618-01, Sigma Aldrich) by centrifugations at 12,000× *g*. Unbound proteins were washed away using PBS 1× and the immunocomplexes were eluted in 1× Laemmli buffer following denaturation at 95 °C for 10 min. Eluted samples were then separated by SDS-PAGE for Western blot analysis.

### 2.7. Antibodies

The following primary antibodies were employed in the immunofluorescence or immunoblotting studies: rabbit polyclonal anti-BECN1 (PA5-96649, Invitrogen), mouse monoclonal anti-Bcl2 (15071S, Cell-Signaling, Danvers, MA, USA), rabbit monoclonal anti-Vps34 (4263S, Cell-Signaling), rabbit polyclonal anti-ATG14 (SAB1306130, Sigma-Aldrich), rabbit monoclonal anti-UVRAG (5320S, Cell-Signaling), rabbit polyclonal anti-AMBRA1 (PA5-88053, Invitrogen), rabbit polyclonal anti-PINK1 (NB100-493, Novus, St. Louis, MO, USA), rabbit polyclonal anti-PRKN (NB100-91921, Novus), rabbit monoclonal anti-BNIP3 (44060S, Cell-Signaling), mouse monoclonal anti-GFP (632381, Clontech), mouse monoclonal anti-GFP (2955S, Cell-Signaling), rabbit polyclonal anti-LC3 (L7543, Sigma-Aldrich), mouse monoclonal anti-Lamp1 (555798, BD Bioscience, Franklin Lakes, NJ, USA), mouse monoclonal anti-p62 (MABC32, Millipore, Burlington, MA, USA), mouse monoclonal anti-β-Actin (A5441, Sigma-Aldrich), mouse monoclonal anti- β-Tubulin (T5201, Sigma-Aldrich).

The following secondary antibodies were employed in the immunoblotting studies: Goat Anti-Rabbit IgG (H + L)-HRP Conjugate (1706515, Bio-Rad), Goat Anti-Mouse IgG (H + L)-HRP Conjugate (1706516, Bio-Rad). The following secondary antibodies or fluorescent dyes were employed in immunofluorescence studies: Alexa Fluor™ plus 488 goat anti-rabbit IgG (A32731, Thermo-Fisher Scientific), Alexa Fluor™ plus 555 goat anti-mouse IgG (A32727, Thermo-Fisher Scientific), Iris™ 5 goat anti-mouse IgG (5WS-07, Molecular Targeting Technologies, Inc., West Chester, PA, USA).

### 2.8. Statistical Analysis

Statistical analysis was performed using GraphPad Prism v8.4.2 software (GraphPad Software, San Diego, CA, USA). Bonferroni’s multiple comparison test was performed after a one-way ANOVA analysis (unpaired, two-tailed). *p*-values < 0.05 were considered significant. All experiments were reproduced at least three times in separate and independent replicates. All data are expressed as mean ± S.D.

## 3. Results

### 3.1. Cloning of BECN1 Transcript Variants from Ovarian Cancer Cells

To investigate the presence of different BECN1 transcript variants in cancer cells, total mRNA was extracted from ovarian cancer NIHOVCAR3 cells. As shown in Figure 1A, the obtained RT-PCR amplicon was of the expected length for a *wild-type* BECN1 transcript (~1981 bp), yet it included more than a single band, which could indicate the presence of either multiple BECN1 transcript variants or of nonspecific PCR products. The RT-PCR product was then digested with the restriction enzymes EcoRI and BamHI and subcloned into the pcDNA 3.1/Zeo (−) vector. The colonies obtained after bacterial transformation were screened by PCR for the presence of the vector carrying BECN1 cDNA using primers designed for annealing the sequences downstream from the RT-PCR primers set. Screening of the colonies revealed one clone carrying the plasmid with a BECN1 cDNA of the expected length for the *wild-type* transcript (Figure 1B, ~765 bp, BECN1-wt) and the other three clones carrying the plasmid with shorter BECN1 cDNA (BECN1-α, -β, and -γ). The identified positive clones were also checked by BglII restriction digestion, and it was confirmed that BECN1 cDNA was inserted in the plasmid in the correct orientation (Appendix A).

To confirm that the short BECN1 isoforms were not nonspecific RT-PCR products, we performed also a nested PCR using the purified RT-PCR product as a template and with the same primers employed for the screening of the colonies. As shown in Figure 1C, the nested PCR confirmed the presence of four different transcript variants, including the isoform corresponding to the *wild-type* BECN1 and the three shorter isoforms (Figure 1C, arrows).

### 3.2. Sequencing Analysis of BECN1 Transcript Variants

Sequence analysis confirmed that we cloned four different BECN1 transcript variants starting from NIHOVCAR 3 total mRNA. The BECN1-wt coding sequence is identical to that of the reference mRNA (GenBank: AF139131) except for one missense point mutation 308C>T resulting in the known amino acid substitution Ala103Val (SNP: VAR_010384).

Multiple sequence alignment with CLUSTALW (European Bioinformatics Institute, EBI) of the nucleotide coding sequences of the clones BECN1-wt, -α, -β, and *-γ* allowed us to investigate the differences between the isoforms (Figure 2A). Compared to BECN1-wt, BECN1-α lacks 143 nt (nucleotides +1141 to +1183). No point mutation was found in the isoform α, indicating that the point mutation 308C>T found in the BECN1-wt is a heterozygous polymorphism. In the isoform BECN1-β, two regions of 229 nt (nucleotides +261 to +489) and 205 nt (nucleotides +920 to +1183) are missing compared to BECN1-wt. Moreover, BECN1-β carries two novel missense point mutations 855A>G and 1244T>C (resulting in the amino acid substitutions Asn211Asp and Ile271Thr, respectively). The isoform BECN1-γ lacks both the region of 229 nt (nucleotides +261 to +489) and the region of 143 nt (nucleotides +1141 to +1183) as in the isoforms BECN1-β and BECN1-α, respectively. Importantly, reviewing the sequences of the isolated isoforms in the database GenBank^®^, we found that the isoform BECN1-α was found also by Niu and colleagues in leukemia cells (GenBank: KC776730) [35], while both BECN1-β and BECN1-γ are novel Beclin isoforms. We submitted the sequences of the novel transcript variants to GenBank^®^ and the isoforms β and γ are now registered under the accession numbers ON805851 and ON805852, respectively.

Next, the deduced amino acid sequences of the isoforms were compared by multiple alignment (CLUSTAL W, Figure 2B). The deletion of 143 nt in BECN1-α alters the reading frame and introduces a premature STOP codon. In the isoforms BECN1-β and BECN1-γ, the deletion of 229 nt results in the removal of 76 amino acids (amino acids 86–163) forming the BH3D and part of the CCD. The second deletion of 205 nt in the isoform BECN1-β results in the loss of additional 68 amino acids but does not introduce a premature STOP codon. The second deletion of 143 nt in the BECN1-γ alters the reading frame and introduces a premature STOP codon as in the isoform BECN1-α. As result, the BECN1 transcript variants translate into protein isoforms of 355 amino acids (BECN1-α), or 306 amino acids (BECN1-β), or 279 amino acids (BECN1-γ). The predicted amino acid sequences were employed to estimate the molecular weight of each isoform using the Compute pI/Mw bioinformatic tool (ExPASy). The estimated molecular weights were 41 kDa for BECN1-α, 35 kDa for BECN1-β, and 32 kDa for BECN1-γ.

The multiple alignments suggest that the missing nucleotide regions found in the 3 short BECN1 isoforms result from alternative splicing. To investigate this hypothesis, the 14,151 nt long BECN1 primary transcript (NCBI Reference Sequence: NC_000017.11) was entered as a template for the prediction of alternative splice sites (ASSP) [37,38]. Putative splicing donor (GT/GC) and acceptor (AG) sites, respectively at the 5′- and 3′-end of the introns, recognized by the software are shown in Figure 2C. Among the recognized splicing sites, we identified the donor and the acceptor sites for the splicing generating the 3 short BECN1 isoforms (Figure 2C, nucleotide sequences in red). As schematized in Figure 2D, the AS results in the skipping of exon 11 (143 nt) in BECN1-α, or exons 5 (91 nt), 6 (137 nt), 10 (61 nt) and 11 (143 nt) in BECN1-β, or exons 5 (91 nt), 6 (137 nt), and 11 (143 nt) in BECN1-γ. Following exon skipping, in the mature RNA, the open reading frame is 1068 nt long for the isoform BECN1-α, 921 nt long for BECN1-β, and 840 nt long for BECN1-γ.

BECN1 cloning and sequencing analysis were performed also starting from the mRNA isolated from the ovarian cancer cell line A2780 as described above for NIHOVCAR3. In A2780, we were able to isolate only the *wild-type* BECN1 transcript and the same isoform BECN1-α found in the NIHOVCAR3 (not shown).

### 3.3. BECN1 Isoforms Show Different Alterations in Their Interactions with VPS34 and BCL2 While Maintaining Their Ability to Bind to ATG14

Based on the predicted amino acid sequence, and as schematized in Figure 3A, the isoforms β and γ, but not α, lack a region of 143 amino acids that includes the BH3D and the initial portion of the CCD of BECN1. For this reason, the isoforms β and γ should fail to interact with BCL2, which binds to the BH3D, and with the autophagy protein AMBRA1, which binds to the amino acids 141–150 [39,40,41]. The isoform β lacks 10 residues of the ECD and therefore may lose its ability to bind with VPS34. The isoform β is also the only isoform that maintains the binding site for VPS15, which is an important component of the mammalian PI3KCIII complex indispensable for the synthesis of PI3P [42,43]. Contrary to the isoform β, both the α and γ isoforms maintain all the amino acids of the ECD but miss the VPS15 binding site as they lack 95 amino acids at their C-termini. Of note, all three isoforms maintain the region of the CCD for the interaction with both ATG14 and UVRAG that are necessary for the formation of either the initiation or maturation autophagy complexes, respectively.

Considering the differences in the protein primary structure among the isoforms, we hypothesized that the short BECN1 isoforms may fail to interact with different autophagy proteins and, as a result, alter autophagy. To investigate their effect on autophagy, each isoform was transiently expressed as BECN1-GFP chimeric protein in cancer cells. The presence of the GFP allowed us to discriminate between the exogenous BECN1 isoform and the endogenous protein. BECN1-GFP isoforms (wt, α, β, or γ) were transiently expressed in the human breast cancer cell line MDA-MB231. MDA-MB231 was chosen to ectopically express the BECN1 isoforms because analysis with COSMIC database (Catalogue Of Somatic Mutations In Cancer; https://cancer.sanger.ac.uk/cosmic, accessed on 24 June 2021) shows that this cell line is genetically categorized for its nonmutant status of the autophagy proteins investigated in our study.

MDA-MB231 cells were transfected with the plasmid carrying the BECN1-GFP *wild-type*, -α, -β, or -γ isoforms and the expression of the exogenous proteins was assessed by immunoblotting after 48 h from the transfection. As estimated by the bioinformatic analysis, each exogenous isoform was found at the expected molecular weight for each BECN1-GFP fusion protein isoform following immunoblotting analysis (Figure 3B). Sub-cellular localization of the isoforms was then assessed by immunofluorescence. Cells were stained for either VPS34 or BCL2 (shown in red in Figure 3C) and exogenous BECN1 isoforms were stained with the anti-GFP antibody. Colocalization between each BECN1 isoforms and either VPS34 or BCL2 was assessed by measuring the intensity of the yellow signal, which results from a close green and red fluorescence. Compared to BECN1-wt, we observed a significant reduction in the colocalization of BECN1-β with VPS34 (Figure 3C, upper panel). This result is consistent with the partial deletion of the ECD observed in this isoform and with the loss of its ability to bind VPS34. Of note, both the isoforms BECN1-α and -γ, which maintain a complete ECD but lack 95 amino acids at their C-terminus, show a slight reduction of colocalization with VPS34 compared to BECN1-wt (although not significant). Compared to BECN1-wt, we also observed a marked reduction of the colocalization of both BECN1-β and -γ with BCL2 (Figure 3C, lower panel). This result is consistent with the lack of the BH3D in both the isoforms. Importantly, as expected by the presence of an intact BH3D, BECN1-α showed colocalization with BCL2 comparable to that of the *wild-type* isoform.

The PI3KCIII complex includes both BECN1 and ATG14 during autophagy initiation, or both BECN1 and UVRAG during autophagosome maturation/vacuolar protein sorting [11,17,18]. To investigate the ability of the isoforms to take part in either PI3KCIII complexes, transfected cells were collected and processed for immunoprecipitation of the exogenous BECN1-GFP. The presence of either ATG14 or UVRAG in the immune complex was checked by immunoblotting. As shown in Figure 3D, we found that only ATG14 is present in the immune complexes of all the BECN1 isoforms. This result indicates that, under steady-state conditions, all the BECN1 isoforms can bind to ATG14, and thus, could retain their ability to initiate autophagy.

Overall, our results show that the C-terminus deletions found in the short BECN1 isoforms α and γ did not significantly alter their ability to interact with VPS34. Interaction with VPS34 is reduced significantly, although not completely impaired, exclusively in the isoform β, where the deletion also includes part of the ECD domain. All the isoforms maintain their ability to bind ATG14, including the isoform β despite its reduced interaction with VPS34. Moreover, our data confirm that deletion of the BH3D impairs BECN1/BCL2 interaction.

### 3.4. BECN1 Isoforms Have Idiosyncratic Effects on Autophagy

To understand whether the isoforms alter autophagy, we investigated the autophagic flux in MDA-MB231 cells transiently transfected with the plasmid carrying BECN1-GFP-*wild-type*, -α, -β, or -γ. To assess the autophagic flux, the cells were exposed to chloroquine (Clq) for the last 4 h. The resulting accumulation of both microtubule-associated protein 1 light chain 3 (MAP1LC3, or LC3) and sequestosome-1 (SQSTM1/p62) was assessed by immunoblotting to evaluate the autophagic flux at basal conditions (Figure 4A). While the cellular level of LC3-II (normalized versus the housekeeping cytoplasmic protein) gives the static picture of the autophagosome and autolysosome present in the cell, the LC3-II/I ratio, measuring the conversion of LC3-I into LC3-II, is a dynamic index of the rate of autophagosome formation [44]. LC3-II/I ratio together with p62/SQSTM1 (which reflects cargo degradation) allow to determine the autophagy flux, that is, the rates of autophagosome formation and degradation [45]. This same analysis performed in the absence and presence of Chloroquine (Clq), which inhibits the latter step, allows to determine how the treatment (in our case the transgenic expression of the BECN1 isoform) affects the autophagy flux [45]. In cells transfected with either the empty vector (Sham) or BECN1-wt, LC3-II/I ratios increased by comparable levels following Clq (~5 times), indicating similar LC3 turnover. Of note, p62 accumulation did not change significantly following Clq in both Sham and BECN1-wt transfected cells, indicating that p62 degradation via autophagy is not significative at basal autophagy conditions. However, it is worth noticing that overexpression of BECN1-wt led to higher p62 levels compared to Sham transfectant. Compared to the control transfectant (BECN1-wt), overexpression of BECN1-α led to a higher LC3-II/I ratio in absence of Clq and reached levels comparable to the control following Clq. As a result, LC3-II/I ratios only doubled following Clq, indicating a reduced LC3 turnover compared to the control. The increased conversion of LC3-I into LC3-II, indicative of autophagosome formation, induced by the ectopic expression of BECN1-α, is evident in the presence of Clq, which inhibits the degradation of autophagosomes. This indicates that newly formed autophagosomes induced by BECN1-α are rapidly degraded, a sign that the autophagic flux is greatly stimulated. This is confirmed by the levels of p62, which were significantly lower in absence of Clq (indicative of cargo degradation) and increased significantly when the cells were incubated with Clq (~5 times). These data suggest high p62 turnover when BECN1-α is overexpressed even if the LC3 turnover is lower than the controls. Overall, these results indicate that the autophagic flux is not impaired when BECN1-α is overexpressed, rather, it is stimulated. Overexpression of BECN1-β also led to a higher LC3-II/I ratio in the absence of Clq compared to the control. However, for this isoform, the LC3-II/I ratio did not increase significantly following Clq, indicating that in this case, the LC3 turnover is significantly impaired. The level of p62 in the absence of Clq was comparable to the control and almost doubled following Clq. Taken together, these results suggest that overexpression of the isoform β could reduce the autophagic flux. Overexpression of BECN1-γ led to LC3-II/I ratios comparable to the control. The level of p62 in the absence of Clq was comparable to the control too, but it increased 2.5 times following Clq. Overall, these results suggest that autophagy is not impaired after overexpression of the isoforms γ. Thus, when overexpressed, the three isoforms differently impact the autophagy process, with the isoform α overstimulating it, the isoform β slightly impairing it, and the isoform γ not much interfering with it.

Next, we assessed the autophagosome–lysosome fusion step of autophagy by immunofluorescence, following ectopic expression of the BECN1 isoforms. Cells were stained for LC3 (green) as a marker of the autophagosome and for the lysosomal associated membrane protein 1 (LAMP1, red) as a marker of the lysosomal compartments. Autolysosomes resulting from autophagosome–lysosome fusion were identified by a yellow signal, resulting from the colocalization of LC3 and LAMP-1. As shown in Figure 4B, overexpression of BECN1-α led to an intense yellow signal comparable to that in BECN1-wt transfectants. On the contrary, overexpression of both β and γ isoforms resulted in a strong reduction of LC3 and LAMP1 colocalization compared to the control, although fusion still occurred. We conclude that compared to the control transfectant (BECN1-wt), autophagosome–lysosome fusion is reduced when the isoforms β and γ are overexpressed, while it is not affected in the case of the isoform α.

### 3.5. BECN1-α Interacts with PRKN Facilitating Mitophagy

BECN1 plays a role also in the induction of mitophagy by promoting the translocation of E3 ubiquitin ligase parkin, or PRKN, onto the mitochondrial membrane [20,46]. To investigate the involvement of BECN1 isoforms in mitophagy, transfected cells were stained for PRKN, and colocalization of BECN1-GFP with PRKN was assessed by immunofluorescence (Figure 5A). Where indicated, cells were treated with 10 μM carbonyl cyanide 3-chlorophenylhydrazone (CCCP), a mitochondrial uncoupling agent that triggers PINK1/PRKN-mediated mitophagy as a stress response mechanism [47]. Measurement of yellow signal intensity showed a significant increase of PRKN/GFP colocalization following treatment with CCCP only when the isoform α was overexpressed. Of note, in both the *wild-type* and isoform γ overexpressing cells, accumulation of PRKN was noticeable following CCCP although no significant increase in PRKN/GFP colocalization was found.

BCL2-interacting protein BNIP3 is an outer mitochondrial membrane protein and a known marker of mitophagy [48,49]. To further investigate the role of isoform α in mitophagy, cells transfected with either the BECN1-GFP *wild-type* or -α were stained for BNIP3 and LC3. LC3/BNIP3 colocalization was assessed by immunofluorescence in either the presence or the absence of CCCP (Figure 5B). Compared to the control, LC3/BNIP3 colocalization seems to increase (although not significantly) already in the absence of CCCP, following overexpression of the isoform α. Treatment with CCCP led to a significant increase of LC3/BNIP3 colocalization only in cells overexpressing the isoform α. These data show that a high amount of mitochondrial *cargo* is routed to the autophagosome for degradation in cells treated with CCCP when the isoform α is overexpressed. On the contrary, overexpression of the *wild-type* isoform showed a slight decrease in LC3/BNIP3 colocalization following CCCP. Additionally, cells transfected as above were stained for LC3 and mitochondria were labelled with MitoTracker Red (MitoRed). MitoRed/LC3 colocalization was assessed in either the presence or the absence of CCCP (Figure 5C). Compared to the control, MitoRed/LC3 colocalization appears to increase (although not significantly) already in the absence of CCCP following overexpression of the isoform α. Treatment with CCCP led to a significant increase of MitoRed/LC3 colocalization only in cells overexpressing the isoform α. These data confirm that more mitochondria are routed to the autophagosome for degradation in cells treated with CCCP when the isoform α is overexpressed.

Next, we checked the protein levels of the mitophagy markers PINK1, PRKN, and BNIP3 by immunoblotting in cells overexpressing the isoform α before and in either the presence or absence of CCCP (Figure 5D) [50]. Following CCCP treatment, we found a significative increase of both PINK1 and PRKN expressions, indicating induction of mitophagy. We checked also AMBRA1 protein levels since AMBRA1 is a BECN1 interactor known to help in the progression of mitophagy [51]. As shown in Figure 5D, AMBRA1 levels also increased significantly following CCCP. Finally, to demonstrate whether mitophagy was induced by BECN1-*α* in a PRKN-dependent manner, we analyzed PRKN/BECN1 interaction by an immunoprecipitation assay (Figure 5E). PRKN was found in the GFP immune complex in both basal conditions and following CCCP treatment. Data show that PRKN binds to the isoform α and confirm the results of the immunofluorescence studies in Figure 5A. Of note, BECN1-*α* binds to both PRKN and ATG14 independently of the exposure to CCCP, while no interaction was found with either PINK1 or AMBRA1. Interaction of BECN1-*α* with both PRKN and ATG14 in the absence of CCCP suggests the sequestration of the large amount of mitochondrial *cargo* when the isoform α is overexpressed. Overall, our data show that the isoform α stimulates mitophagy.

## 4. Discussions

Autophagy-related genes undergo alternative splicing with the generation of different protein isoforms known to regulate autophagy [34]. Two shorter isoforms of BECN1 have been found respectively in leukemia and HeLa cells. The first isoform lacks exon 11 and reduces the induction of autophagy following starvation [35]. The second short isoform lacks both the exons 10 and 11, and it has been reported to induce mitophagy [36]. Considering the pivotal role of BECN1 in the regulation of autophagy and its importance in cancer, more work is currently needed to understand the BECN1 splicing isoforms present in different cancers and how each isoform may affect autophagy.

Here, we expand on the previous studies by investigating the BECN1 isoforms isolated from ovarian cancer cells. Our results add to the previous studies by confirming the presence also in ovarian cancer cells A2780 and NIHOVCAR3 of the short BECN1 isoforms lacking exon 11 (BECN1-α) already found by Niu and colleagues in leukemia cells [35]. Moreover, we were able to identify two novel short isoforms lacking exons 5, 6, 10, and 11 (BECN1-β), or exons 5, 6, and 11 (BECN1-γ). These two novel isoforms β and γ were found exclusively in NIHOVCAR3. In both A2780 and NIHOVCAR3, the short BECN1 isoforms were found to be present along with the canonical *wild-type* BECN1 isoform. Sequencing analysis showed the presence of a known missense point mutation 308C>T (SNP: VAR_010384) in the BECN1-wt isoform of NIHOVCAR3. Moreover, two novel missense point mutations 855A>G and 1244T>C were found in the isoform BECN1-β. In A2780, no missense point mutations were found in BECN1-wt and BECN1-α carries a novel missense point mutation 788A>G (not shown). Together with the previous works, our data support the notion that accelerated RNA splicing events in tumors can generate cancer-specific variants [33], and that variable expression of BECN1 isoforms can be found in different tumor cells and need to be investigated.

We assessed the role on autophagy of each isoform isolated form NIHOVCAR3 (BECN1-α, -β, and -γ) by exogenously expressing them in MDA-MB231 cells. As schematized in Figure 4A, the three short isoforms lack different domains of BECN1. The isoform α has only one deletion of 143 nt that alters the reading frame and introduces a premature STOP codon with the resulting loss of 96 amino acids at its C-terminal. The C-terminal deletion in BECN1-α includes the BARA domain (amino acids 425–450) necessary for the binding with VPS15. In cells overexpressing BECN1-GFP-α, and under basal autophagy conditions, we found that the exogenous isoform maintained its ability to interact with VPS34, BCL2 and to bind with ATG14 (Figure 3). These results are consistent with the presence of intact ECD, BH3D, and CCD domains in this isoform. LC3 turnover assay following Clq (Figure 4A) showed that cells overexpressing BECN1-α have a reduced LC3 turnover compared the control transfectant overexpressing the *wild-type* isoform. However, autolysosome analysis confirmed that autophagosome–lysosome fusion occurs as well as in the control transfectant (Figure 4B). Of note, p62 levels increased significantly following Clq compared to the control (Figure 4A). Overall, these results indicate that the autophagic flux is not impaired when BECN1-α is overexpressed, rather, it is stimulated. Compared to the control and the other short isoforms β and γ, BECN1-α showed also increased interaction with PRKN, either in the absence or presence of CCCP (Figure 5A). BNIP3/LC3 and MitoRed/LC3 staining showed a significative increase in the number of mitochondria colocalizing with LC3 following overexpression of BECN1-α and CCCP treatment compared to the control (Figure 5B,C), indicating that a large portion of mitochondria was routed for autophagy degradation. In cells overexpressing BECN1-α, protein levels of mitophagy markers PINK1, PRKN, BNIP3 and of AMBRA1 increased following CCCP treatment (Figure 5D). Actual binding between BECN1-α and PRKN was confirmed by co-immunoprecipitation assay either in the absence or in the presence of CCCP (Figure 5E). We conclude that BECN1-α stimulates mitophagy. Niu and colleagues, who previously described the same isoform in leukemia cells [35], concluded that this isoform reduces the induction of autophagy following starvation and did not report stimulation of mitophagy. Interestingly, mitophagy induction has been observed by Chen and colleagues with the isoform lacking both the exons 10 and 11 [36]. Importantly, our analysis was performed in full media and by overexpressing the isoform α in the presence of the endogenous *wild-type* BECN1 of MDA-MB231 cells. It is likely that the effect of BECN1-α on autophagy depends on the cellular context and the experimental conditions.

The isoform β lacks the first region of 229 nt at its N-terminus resulting in the loss of 76 amino acids forming the BH3D and part of the CCD including the AMBRA1 binding site. This isoform lacks also the second region of 205 nt at its C-terminus which results in the loss of 68 amino acids that includes part of the ECD. Of note, this second deletion does not introduce a premature STOP codon and, as a result, the VPS15 binding site is present. After overexpression of BECN1-GFP-β, we found that the exogenous isoform lost its ability to interact with both VPS34 and BCL2 under basal conditions (Figure 3A,B). This result is consistent with the partial loss of the ECD and with the loss of the BH3D. On the contrary, and as expected from the presence of an intact binding site for ATG14/UVRAG in the CCD domain, BECN1-β maintained its ability to bind with ATG14 (Figure 3C). Although p62 levels increased following Clq compared to the control, the LC3 turnover assay clearly showed that overexpression of BECN1-β lead to a strong reduction of autophagy (Figure 4A). Autolysosome analysis confirmed that autophagosome–lysosome fusion is reduced compared to the control transfectant (Figure 4B). Overall, these results are consistent with the absence of part of the ECD and a resulting loss of the VPS34 binding site and show that overexpression of this isoform reduces autophagy with a dominant negative effect over the endogenous *wild-type* BECN1.

The isoform γ lacks both the same region of 143 nt as the isoform α and the same region of 229 nt as the isoform β. As a result, γ lacks the BH3D, part of the CCD including the AMBRA1 binding site and the C-terminal region within the BARA domain. Overexpression of BECN1-GFP-γ shows that the exogenous isoform maintains its ability to interact with VPS34, but loses its ability to interact with BCL2 (Figure 3A,B). This result is consistent with the presence of an intact ECD and with the loss of BH3D. As expected from the presence of an intact binding site for ATG14/UVRAG in the CCD domain, BECN1-γ maintained its ability to bind with ATG14 (Figure 3C). LC3 and p62 turnover assay showed that overexpression of BECN1-γ has little effect on autophagy (Figure 4A). However, autolysosome analysis showed that autophagosome–lysosome fusion is reduced compared to the control transfectant (Figure 4B). We speculate that the ability of this isoform to bind with VPS34 is enough to not compromise autophagy. It is possible that the presence of the endogenous *wild-type* BECN1 could overcome any inhibitory effect of the γ isoform on autophagy.

As mentioned above, the effects on autophagy observed for each isoform could depend on the genetic context of the cell line and the experimental conditions. For these reasons, we cannot rule out the possibility that under starvation, or in a different cell line, or in the absence of the *wild-type* BECN1, these isoforms could show a different impact on autophagy therefore, more investigation is needed to fully understand their role in autophagy. For example, a recent study has reported a mutant BECN1 harboring a non-functional BH3D that can eventually inherit a gain of function and enhances the autophagic flux regardless of nutrient abundance [52]. Therefore, we hypothesize that the isoforms β and γ could bypass the negative regulation by BH3 proteins and the upstream stress signaling cascades and upregulate autophagy under starvation or hypoxic conditions. This includes mutations in BECN1 interactors that might have a different impact on the isoforms and consequently on autophagy. For instance, p53 has been shown to interact (through the BH3D) and direct the proteasomal degradation of BECN1 which in turn downregulates autophagy [53]. Therefore, mutated p53 that cannot bind to BECN1 or a *wild-type* p53 in combination with BECN1 isoforms lacking the BH3D may result in upregulation of autophagy. Indeed, mutated p53 along with low BECN1 mRNA expressions significantly correlate with a better prognosis associated with upregulated autophagy and sensitization to cis-platinum in ovarian cancer patients [9]. In conclusion, it is important to consider that different BECN1 variants can compete with the *wild-type* for binding with the interactors depending on both the microenvironmental stimuli and the genetic context leading to different effects on autophagy and cancer patient prognosis.

The limitations of our study are the following: (i) The short isoforms were isolated by using RT-PCR primer sets designed to clone the full length canonical *wild-type* BECN1 isoform. Thus, we cannot exclude that other short isoforms could be present in the same cell lines; (ii) The splicing variants here described were obtained only from two ovarian cancer cell lines. BECN1 splicing isoforms have been reported in other cell lines [35,36]. We cannot exclude that these isoforms result from specific stresses or molecular background of these cancer cell lines and that such AS can occur in actual tumors. However, since the splicing consensus sequences (as we have shown here) exist in the BECN1 gene and the splicing machinery is ubiquitous, it is likely that, though with variable level of expression and depending on the environmental stimuli, these splicing isoforms can be produced by other types of cancer cells. Clearly, this has to be proven experimentally; (iii) BECN1 is involved in several cellular processes in addition to autophagy, including apoptosis and metabolism regulation. Moreover, autophagy-independent roles of BECN1 have been reported, including autophagy-independent antitumoral effects [54]. In the present study, we investigated the role of the identified isoforms on autophagy. More investigation is needed to understand whether the short BECN1 isoforms maintain their functional role in other cellular processes or have any impact on cancer; (iv) BECN1-independent autophagy can occur [55]. Thus, we cannot exclude that autophagy may be able to overcome the presence of defective BECN1 isoforms in different contexts.

In conclusion, our study adds evidence to the existence of different BECN1 isoforms in cancer and shows that their presence can alter autophagy in distinctive ways.

At this stage, we cannot clarify the biological significance of each isoform and whether they could serve as clinical biomarkers, yet we hypothesize that such isoforms emerge under particular environmental stress conditions such as starvation or hypoxia, or possibly in cancer stem cells. These hypotheses need to be tested. In the future, we aim to investigate each isoform in 3D cell models under induced hypoxic and starvation conditions and the presence of cytokines and growth factors to better mimic the solid tumor micro-environment. Considering the impact that short isoforms could have on cancer, we will also consider the investigation of pharmacological approaches to prevent the formation or neutralize the function of a specific short BECN1 isoform.

## Figures and Tables

**Figure 1 biomolecules-12-01069-f001:**
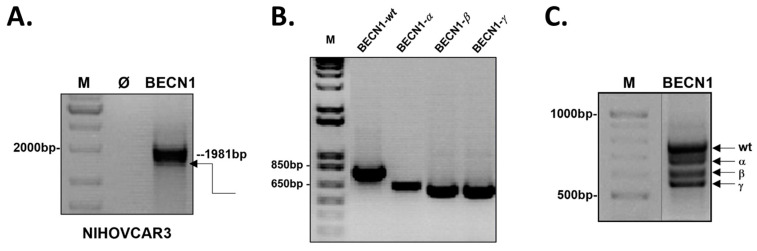
Isolation of BECN1 transcript variants from ovarian cancer cells NIHOVCAR3. Agarose gel electrophoresis of the products of (**A**) the RT-PCR, (**B**) screening PCR, and (**C**) nested PCR. Expected bands for the *wild-type* BECN1 isoform are indicated ((**A**,**B**), dotted lines). Arrow in (**A**) indicates putative additional RT-PCR product. Arrows in (**C**) indicate the *wild-type* BECN1 (wt) and the short BECN1 isoforms α, β, and γ. Molecular weight standard, M. Negative control (no template), Ø.

**Figure 2 biomolecules-12-01069-f002:**
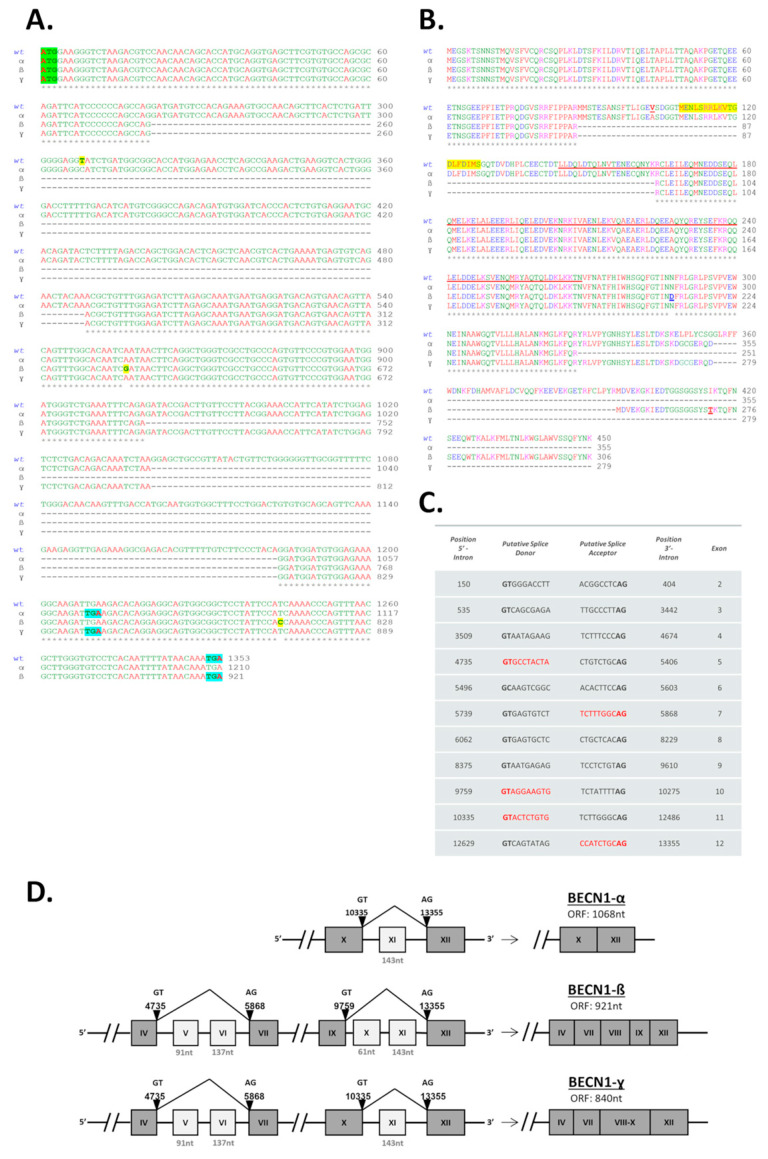
Sequencing analysis of BECN1 transcript variants. (**A**) Multiple sequence alignment of the nucleotide coding sequences of the BECN1 isoforms (wt, α, β, and γ) isolated from NIHOVCAR3. Nucleotide numbering: nucleotide +1 is the A of the ATG-translation initiation codon in the BECN1-wt mRNA. ATG-translation initiation codons are in green. TGA stop codons are in blue. Missense point mutations (308C>T, 855A>G, and 1244T>C) found in the isoforms BECN1-wt and -β are in yellow. (**B**) Multiple sequence alignment of the amino acid sequences of the BECN1 isoforms (wt, α, β, and γ). Amino acids 108–127 of the BH3D domain are highlighted in yellow. Amino acids 144–269 of the CCD domain are underlined in red. Substitutions Asn211Asp and Ile271Thr resulting from the missense point mutations found in the isoform BECN1-β are in underlined bold letters. (**C**) Putative splicing donor (GT/GC) and acceptor (AG) sites recognized by the software ASSP. Nucleotide sequences of the donor/acceptor site for the splicing generating the 3 short BECN1 isoforms are in red. Nucleotide numbering: Nucleotide +1 is the first nucleotide of the primary transcript (NCBI Reference Sequence: NC_000017.11). Nucleotide positions define splice site boundaries (nucleotide position at 5 and 3′ introns). (**D**) Schematic representation of the exon skipping resulting from the alternative splicing of the BECN1 isoforms α, β, and γ. Skipped exons are in light gray. Nucleotide positions of the splicing donor/accepter involved in the alternative splicing are indicated (arrowheads).

**Figure 3 biomolecules-12-01069-f003:**
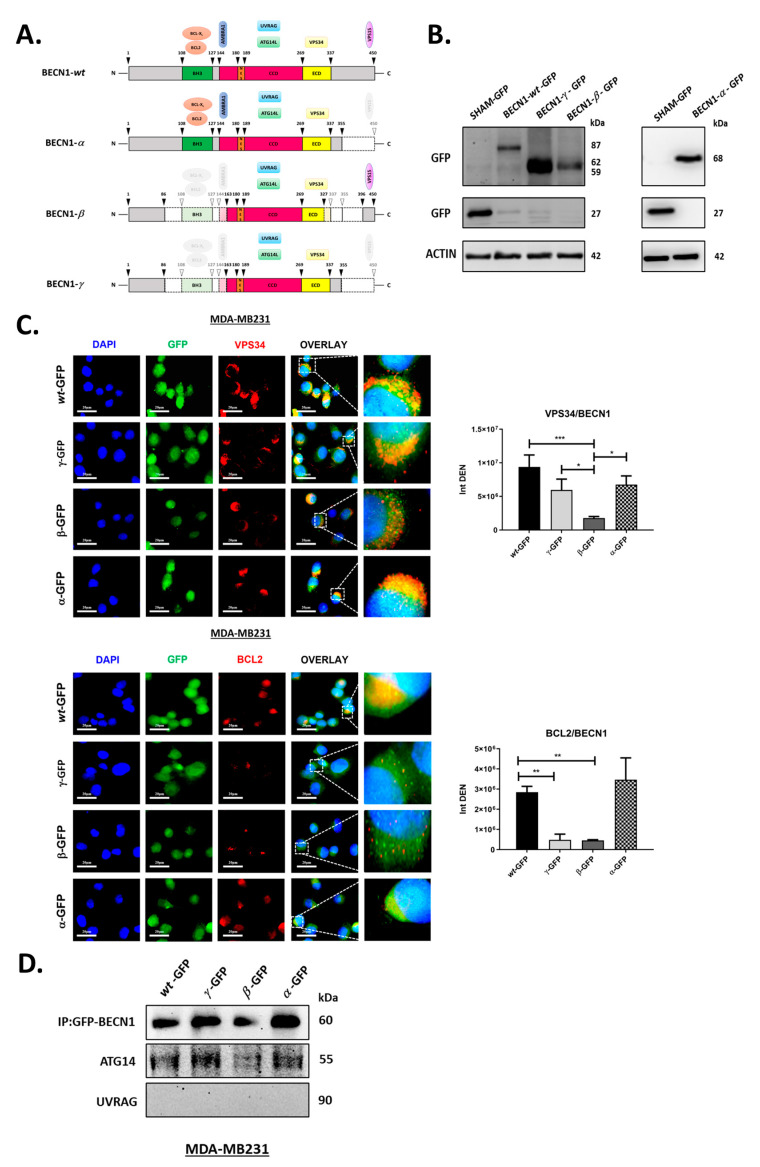
BECN1 isoforms show different alterations in their interactions with VPS34 and BCL2 while maintaining their ability to bind to ATG14. (**A**) Schematic representation of BECN1 isoforms primary protein structure and domains. BECN1 domains and interacting proteins investigated in this study are shown. Arrowheads indicate the positions of the amino acids flanking BECN1 domains, or of the deletions found in the short isoforms. (**B**) MDA-MB231 cells were transfected with the empty pEGFP-N1 vector (SHAM-GFP) or with the vector carrying BECN1-wt-GFP, BECN1-α-GFP, BECN1-β-GFP, or BECN1-γ-GFP. After 48 h, cells were harvested and processed for immunoblotting with anti-GFP antibody. Molecular weights of the bands expected for GFP or for each BECN1 isoform fused with the GFP are shown. (**C**) MDA-MB231 cells plated on coverslips and transfected with pEGFP-N1 vector carrying BECN1-wt-GFP (wt-GFP), BECN1-α-GFP(α-GFP), BECN1-β-GFP (β-GFP), or BECN1-γ-GFP (γ-GFP). After 48 h, cells were fixed, stained for either VPS34 (upper panel, red) or BCL2 (lower panel, red) and imaged by fluorescence microscopy. GFP fluorescence (green) labels the BECN1-GFP fusion proteins. Nuclei were stained with DAPI (blue). Scale bars: 20 μm. Histograms show the intensity densities of yellow signals (Int DEN), which result from close green and red fluorescence and are representative of the level of colocalization between BECN1/VPS34 or BECN1/BCL2. Asterisks indicate significantly different yellow signal intensities (* *p* < 0.05, ** *p* < 0.01, *** *p* < 0.001). Error bars, SD. (**D**) MDA-MB231 cells were plated on Petri dishes and transfected as in (**C**). After 48 h, cells were processed for the immunoprecipitation of BECN1-GFP isoforms as described in the Materials and Methods section. Immune complexes were separated by SDS-PAGE and processed for immunoblotting with the indicated antibodies.

**Figure 4 biomolecules-12-01069-f004:**
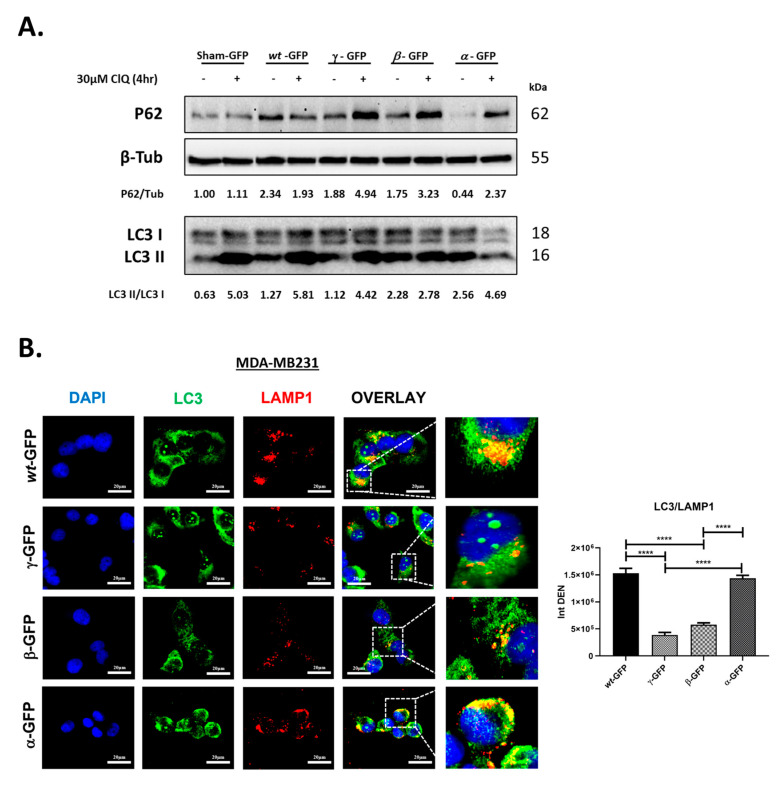
BECN1 isoforms have idiosyncratic effects on autophagy. (**A**) MDA-MB231 cells were transfected with pEGFP-N1 empty vector (Sham-GFP) or the vector carrying BECN1-wt-GFP (wt-GFP), BECN1-α-GFP(α-GFP), BECN1-β-GFP (β-GFP), or BECN1-γ-GFP (γ-GFP). After 48 h, cells were harvested and processed for immunoblotting with the indicated antibodies. Where indicated, cells were exposed to 30 µM Clq for the last 4 h. Band intensities were determined by densitometric analysis and LC3-II/I or p62/tubulin ratios are shown. (**B**) MDA-MB231 cells plated on coverslips were transfected as in (**A**). After 48 h, cells were fixed, stained for LAMP1 (red) and LC3 (green), and imaged by fluorescence microscopy. Nuclei were stained with DAPI (blue). Cells shown in the panels were all positive for GFP fluorescence, indicating that all the cells were expressing the exogenous isoforms. Scale bars: 20 μm. Histograms show the intensity densities of yellow signals (Int DEN), which result from close green and red fluorescence and are representative of the level of colocalization between LAMP and LC3. Asterisks indicate significantly different yellow signal intensities (**** *p* < 0.0001). Error bars, SD.

**Figure 5 biomolecules-12-01069-f005:**
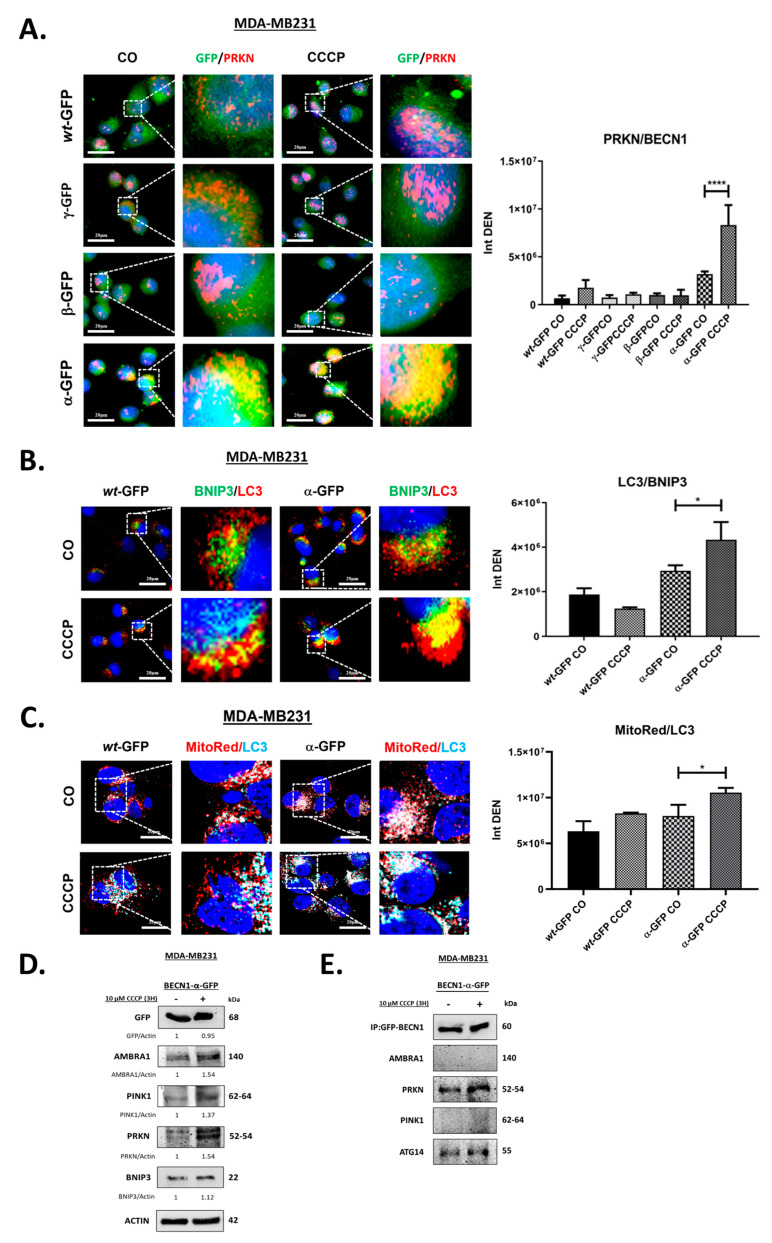
BECN1-*α* interacts with PRKN stimulating mitophagy. (**A**) MDA-MB231 cells plated on coverslips were transfected with pEGFP-N1 vector carrying BECN1-wt-GFP (wt-GFP), BECN1-α-GFP (α-GFP), BECN1-β-GFP (β-GFP), or BECN1-γ-GFP (γ-GFP). After 48 h, cells were fixed, stained for PRKN (red) and imaged by florescence microscopy. Where indicated, cells were treated with 10 μM CCCP for the last 3 h. Nuclei were stained with DAPI (blue). Scale bars: 20 μm. Histograms show the intensity densities of yellow signals (Int DEN), which result from close green and red fluorescence and are representative of the level of colocalization between PRKN and GFP. Asterisks indicate significantly different yellow signal intensity (**** *p* < 0.0001). Error bars, SD. (**B**) MDA-MB231 were plated on coverslips and transfected with pEGFP-N1 vector carrying BECN1-wt-GFP (wt-GFP) or BECN1-α-GFP (α-GFP). After 48 h, cells were fixed, stained for LC3 (red) and BNIP3 (green), and imaged by fluorescence microscopy. Where indicated, cells were treated with 10 μM CCCP for the last 3 h. Nuclei were stained with DAPI (blue). Scale bars: 20 μm. Histograms show the intensity densities of yellow signals (Int DEN), which result from close green and red fluorescence and are representative of the level of colocalization between BNIP3 and LC3. Asterisks indicate significantly different yellow signal intensity (* *p* < 0.05). Error bars, SD. (**C**) MDA-MB231 were plated on coverslips and transfected and treated as in (B). After 48 h, cells were incubated with 500 nM Mitotracker™ RED (MitoRed) for the last 15 min at 37 °C. Following Mitotracker incubation, cells were washed with PBS, fixed, stained for LC3 (cyan), and imaged by fluorescence microscopy. Nuclei were stained with DAPI (blue). Scale bars: 20 μm. Histograms show the intensity densities of white signals (Int DEN), which result from close red and cyan fluorescence and are representative of the level of colocalization between Mitotracker and LC3. Asterisks indicate significantly different white signal intensity (* *p* < 0.05). Error bars, SD. (**D**) MDA-MB231 cells were plated on Petri dishes and transfected with pEGFP-N1 vector carrying BECN1-α-GFP (α-GFP). After 48 h, cells were harvested and processed for immunoblotting with the indicated antibodies. Where indicated, cells were treated with 10 μM CCCP for the last 3 h. Band intensities were determined by densitometric analysis and GFP/Actin, AMBRA1/Actin, PINK1/Actin, PRKN/Actin, BNIP3/Actin ratios are shown. (**E**) Cell homogenates from (**C**) were processed for the immunoprecipitation of BECN1-α-GFP isoform, as described in the Materials and Methods section. Immune complexes were separated by SDS-PAGE and processed for immunoblotting with the indicated antibodies.

## Data Availability

Not applicable.

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
