# Peer review of "Isolation, Characterization, and Autophagy Function of BECN1-Splicing Isoforms in Cancer Cells"

_biomolecules, 2022, doi:10.3390/biom12081069_

Round 1

Reviewer 1 Report

The present reviewed manuscript is both interesting and important since it brings scientific evidence although not robust about beclin-1 splice variants, thereby suggesting another way some malignant cells might influence the process of their survival via autophagy. The form, concept and experiments as well as the interpretation of acquired data are adequate thus it is recommended that only relatively minor changes are made to make the manuscript published.

1. the greatest concern is over the robustness of the presented findings. The fact that splice variants were found in just one ovarian cancer cell line with another breast cancer cell line as a control seems to be just very preliminary. Authors must at least introduce this aspect into their discussion.  For instance, is is possible that the discovered splice variants might correlate with the specific molecular background of the particular cancer cell line? Is is reasonable to expect that this phenomenon would be restricted to some kinds of cancer or be more prevalent and universal? Cancer cell lines are standardly used for research but often when the data from them are correlated with the ones from primary cultures obtained from patients, differences are found. These facts must be considered, duly discussed if not experimentally verified.

2. minor issues

 - abstract - all abbreviations need to be explained 

- Fig. 1 individual panels should be labeled A, B, C, in the panel C instead of providing the number 765 bp the symbol wt should be used 

- used cell lines - at which passages they were used?

- retrotranscribed should be replaced with reversaly transcribed 

magnifications should be added into legends of figures showing microscopic images

Author Response

The present reviewed manuscript is both interesting and important since it brings scientific evidence although not robust about beclin-1 splice variants, thereby suggesting another way some malignant cells might influence the process of their survival via autophagy. The form, concept and experiments as well as the interpretation of acquired data are adequate thus it is recommended that only relatively minor changes are made to make the manuscript published.

We appreciate the generous comments of the reviewer about our manuscript. Indeed, our work is adding to the current knowledge on BECN1 splicing variants in cancer cell lines and explore their functional role on autophagy. Although further investigation is needed to fully understand the biological significance of the identified BECN1 isoforms in actual tumors, we hope that our work will provide useful and novel information to the field.

  1. the greatest concern is over the robustness of the presented findings. The fact that splice variants were found in just one ovarian cancer cell line with another breast cancer cell line as a control seems to be just very preliminary. Authors must at least introduce this aspect into their discussion.  For instance, is is possible that the discovered splice variants might correlate with the specific molecular background of the particular cancer cell line? Is is reasonable to expect that this phenomenon would be restricted to some kinds of cancer or be more prevalent and universal? Cancer cell lines are standardly used for research but often when the data from them are correlated with the ones from primary cultures obtained from patients, differences are found. These facts must be considered, duly discussed if not experimentally verified.

We agree with this important observation of the reviewer. As mentioned in the introduction, the literature (two papers) reports the existence of Beclin-1 splicing isoforms.  Indeed, since the splicing consensus sequences (as we have shown here) exist in the BECN1 gene and the splicing machinery is ubiquitous, it is likely that, though with variable level of expression and depending on the environmental stimuli, these splicing isoforms can be produced by other types of cancer cells. However, we agree that more investigation is needed to confirm that such isoforms are expressed also in other cancer cell lines and in actual tumors. We have now included the observations of the reviewers among the limitations of our work (Discussion; lines 697-722).

  1. minor issues

 - abstract - all abbreviations need to be explained 

We have now explained/removed all the abbreviations in the abstract.

- Fig. 1 individual panels should be labeled A, B, C, in the panel C instead of providing the number 765 bp the symbol wt should be used

We thank the reviewer for pointing out these issues in Fig.1. We have now updated Figure 1 as indicated by the reviewer.

- used cell lines - at which passages they were used?

We have now included the number of passages of cell lines in the Materials and Methods (Subsection 2.1. Cell Cultures; line 113).

- retrotranscribed should be replaced with reversaly transcribed 

We agree with the reviewer, and we have now replaced “retrotranscribed” with “reversely transcribed”.

magnifications should be added into legends of figures showing microscopic images

We have now provided the objective magnification strength in the Materials and Methods (Subsection 2.5. Immunofluorescence; line 198). Please note that since the same magnification was used for all the images shown in the paper, we have added this information only once in the Materials and Methods.

Reviewer 2 Report

In this article the authors report the effect of different isoforms of BECN1 on autophagy regulation. The article is well written, and the topic is very interesting. Only few studies have been focused on BECN1 post-transcriptional regulation , and this report may serve as input for the understanding the regulation of other genes. 

However, I have some points that should be addressed:

- The role of autophagy in cancer is highly controversial. The authors should mention this aspect in the introduction, specifing the lysosomal regulation by TFEB 10.3390/cells10102752, 10.3390/ijms22010179.

- It would be important to clarify what is the biological significance of BECN1 isoforms. Why they exists? Are there any stresses that induce differentially these isoforms? Why these isoforms are expressed although their effect on autophagy is limited ? In what tissue are expressed? Could serves as clinical biomarkers? please, address to these question. 

- I kindly remind to the authors that there are many proteins that although have lost the original function, they are able to interact with other proteins to modulate some signaling pathways. In this case, it deserve to be mentioned that BCN1 is a multifunctional protein that plays roles in many processes, as apoptosis, regulation of metabolism ect 10.1210/jc.2012-2679; 10.18632/oncotarget.2265, .  Hence, the effect of these isoforms could be discussed to which concern others processes.

-Figure 4A: LC-3 II should be normalized to tubulin, instead than LC-3 I

-The authors state that interaction of BCN1 alpha with PRKN leads mitophagy. I have some doubts about this point. How the authors addressed that mitophagy is induced? mitochondria staining (mitotracker) should be performed after BCN1 alpha  expression. Other markers related to mitophagy are parkin/PINK1 ect... the authors could evaluate one of those markers to be sure that mitophagy is induced.

- Finally, a rescue experiment consisting in inhibiting the interaction of BCN1-alpha/PRKN with a specific siRNA against PRKN and then analyzing mitophagy should be performed to provide more proofs for the BCN1 alpha- induced mitophagy

Author Response

Reviewer #2:
In this article the authors report the effect of different isoforms of BECN1 on autophagy regulation. The article is well written, and the topic is very interesting. Only few studies have been focused on BECN1 post-transcriptional regulation , and this report may serve as input for the understanding the regulation of other genes.

We thank the reviewer for the positive comments about our manuscript. We agree that there are still few studies investigating BECN1 alternative splicing isoforms and their role on autophagy. We hope that our findings will add novel information to the field and will prompt further investigation on this topic.

However, I have some points that should be addressed:

- The role of autophagy in cancer is highly controversial. The authors should mention this aspect in the introduction, specifing the lysosomal regulation by TFEB 10.3390/cells10102752, 10.3390/ijms22010179.

We agree with the reviewer, and we have now included in the introduction a specific section explaining the controversial dual role of autophagy in cancer (Introduction; lines 63-74). We also discussed the importance of TFEB in the regulation of the lysosomal compartments and how this important gene, along with BECN1, can play both a tumor-suppressor or -promoter role.

- It would be important to clarify what is the biological significance of BECN1 isoforms. Why they exists? Are there any stresses that induce differentially these isoforms? Why these isoforms are expressed although their effect on autophagy is limited ? In what tissue are expressed? Could serves as clinical biomarkers? please, address to these question. 

- I kindly remind to the authors that there are many proteins that although have lost the original function, they are able to interact with other proteins to modulate some signaling pathways. In this case, it deserve to be mentioned that BCN1 is a multifunctional protein that plays roles in many processes, as apoptosis, regulation of metabolism ect 10.1210/jc.2012-2679; 10.18632/oncotarget.2265.  Hence, the effect of these isoforms could be discussed to which concern others processes.

We agree and thank the reviewer for these two important observations. In the limitation section of the revised manuscript (Discussion; lines 697-722), we have now discussed: (a) The putative biological significance of the BECN1 isoforms; (b) The possibility that the identified short isoforms could maintain their functional role in other cellular processes not investigated in our study; (c) The possibility that autophagy may overcome the presence of defective BECN1 isoforms (Beclin 1-independent autophagy).

-Figure 4A: LC-3 II should be normalized to tubulin, instead than LC-3 I

We understand that the reviewer would like to have a homogeneous presentation of the protein expression according to the standard way of normalizing it versus the housekeeping protein. However, in the case of proteins functionally linked such as LC3-I and LC3-II (and as it is the case for a phospho-protein and its unphosphorylated cognate) it is more relevant to determine their relative ratio.

This is clearly stated in the Guidelines for assessing autophagy (Klionsky DJ et al., 2021), to which both the corresponding authors (CF and CI) of this manuscript have contributed. In brief, as we also report in the text “While the cellular level of LC3-II (normalized versus the housekeeping cytoplasmic protein) gives the static picture of the autophagosome and autolysosome present in the cell, the LC3-II/I ratio, measuring the conversion of LC3-I into LC3-II, is a dynamic index of the rate of autophagosome formation (Mizushima N and Yoshimori T, 2007). LC3-II/I ratio together with p62/SQSTM1 (which reflects cargo degradation) allow to determine the autophagy flux, that is the rates of autophagosome formation and degradation (Klionsky DJ et al., 2021).  This same analysis performed in the absence and presence of Chloroquine (Clq), which inhibits the latter step, allows to determine how the treatment (in our case the transgenic expression of the BECLIN isoform) affects the autophagy flux (Klionsky DJ et al., 2021).”

To make it clear for the reviewer, we report here the densitometry of LC3-II/tubulin.

Please, note that due to technical issues, the last lane was under-developed for LC3 staining, so that the LC3-II/tubulin ratio is misleading. In fact, it would be inconsistent with the data on p62. However, in this case the ratio LC3-II/LC3-I is not affected.

-The authors state that interaction of BCN1 alpha with PRKN leads mitophagy. I have some doubts about this point. How the authors addressed that mitophagy is induced? mitochondria staining (mitotracker) should be performed after BCN1 alpha  expression. Other markers related to mitophagy are parkin/PINK1 ect... the authors could evaluate one of those markers to be sure that mitophagy is induced.

We agree with the reviewer concerns about the stimulation of mitophagy by the BECN1-a isoform. As suggested by this reviewer, we have now performed a Mitotracker/LC3 staining following the overexpression of both BECN1-wt and BECN1-a and in the presence or absence of CCCP (Figure 5C). This experiment shows a significative increase of mitochondria colocalizing with LC3 following overexpression of a isoform and CCCP treatment compared to the control. This new data confirms our previous BNIP3/LC3 staining (Figure 5B) and indicates an increase of mitochondria routed for autophagy degradation. Moreover, we assessed the protein levels of the mitophagy markers PINK1, PRKN and BNIP3 by immunoblotting following the overexpression of BECN1-a and in either the presence or absence of CCCP (Figure 5D). Additionally, in the same homogenates, we assessed the protein levels of AMBRA1 since this BECN1 interactor is known to help the progression of mitophagy (Humbeeck CV et al., 2011) (Figure 5D). The results of our new immunoblotting confirmed increased expression of PINK1, PRKN and AMBRA1 following CCCP. Overall, our new data confirm that the isoform BECN1-a stimulates mitophagy.

The new results of Figure 5 are now discussed in the subsection 3.5. of the Results (lines 532-546) and in the Discussion (lines 630-635). The experimental procedure of the Mitotracker staining is detailed in the Materials and Methods (2.5. Immunofluorescence; lines 205-210).

The image below shows the efficiency of the transfection of BECN1-α-GFP.

- Finally, a rescue experiment consisting in inhibiting the interaction of BCN1-alpha/PRKN with a specific siRNA against PRKN and then analyzing mitophagy should be performed to provide more proofs for the BCN1 alpha- induced mitophagy

Although we agree with the reviewer comment, we were not able to perform the suggested experiment. In fact, cells were already transiently transfected with each BECN1 isoform and further siRNA transfection resulted in cell death. In the future, a successful generation of stable transfectants for this BECN1 isoform shall allow to perform the experiment suggest by the reviewer.

Reviewer 3 Report

The manuscript “Isolation, Characterization and Autophagy Function of BECN1-Splicing Isoforms in Cancer Cells” represents an original study on alternative splicing of Beclin 1, known as an autophagy regulator and tumour suppressor gene. Authors have described the isolation, molecular structure and functional characteristics of three transcript variants (alfa, beta and gamma, compared to wild type) of Beclin 1 in cultured human cancer cells. Three cell lines have been used in different stages of the study, including two cell lines of ovarian cancer and one line, representing triple negative breast cancer. Authors have reported the structural differences between alfa, beta and gamma variants compared to the wild-type transcript, as well as the distinctive influences on autophagy and mitophagy. The study is carefully designed. The introduction provides logical, detailed and interesting basic information on Beclin 1, its role in autophagy and mitophagy as well as on alternative splicing. The materials and methods are thoroughly described. The results are explicitly presented and supplemented by outstanding figures. In the discussion, authors have interpreted their findings and assessed them in the light of preceding research; both the novelty and limitations of the study have been analysed. Despite the accurate study design, a general major drawback remains as the molecular processes in the whole cancer tissues (including the microenvironment and subjected to dynamic temporal alterations) can differ from the events in artificially structured experimental environment. However, authors have recognized this aspect and accounted for it outlining the directions for future research.

The information represented in the article can undoubtedly be useful for future studies thus benefitting the development of medical science. Considering the appropriate scientific level of the study, I recommend to accept it for the publication in the current form.

Author Response

Reviewer #3:
The manuscript “Isolation, Characterization and Autophagy Function of BECN1-Splicing Isoforms in Cancer Cells” represents an original study on alternative splicing of Beclin 1, known as an autophagy regulator and tumour suppressor gene. Authors have described the isolation, molecular structure and functional characteristics of three transcript variants (alfa, beta and gamma, compared to wild type) of Beclin 1 in cultured human cancer cells. Three cell lines have been used in different stages of the study, including two cell lines of ovarian cancer and one line, representing triple negative breast cancer. Authors have reported the structural differences between alfa, beta and gamma variants compared to the wild-type transcript, as well as the distinctive influences on autophagy and mitophagy. The study is carefully designed. The introduction provides logical, detailed and interesting basic information on Beclin 1, its role in autophagy and mitophagy as well as on alternative splicing. The materials and methods are thoroughly described. The results are explicitly presented and supplemented by outstanding figures. In the discussion, authors have interpreted their findings and assessed them in the light of preceding research; both the novelty and limitations of the study have been analysed. Despite the accurate study design, a general major drawback remains as the molecular processes in the whole cancer tissues (including the microenvironment and subjected to dynamic temporal alterations) can differ from the events in artificially structured experimental environment. However, authors have recognized this aspect and accounted for it outlining the directions for future research.

The information represented in the article can undoubtedly be useful for future studies thus benefitting the development of medical science. Considering the appropriate scientific level of the study, I recommend to accept it for the publication in the current form.

We appreciate the generous comments of the reviewer and share the same optimism regarding the future applications of our findings. As noted by the reviewer, we indeed recognized the limitation of investigating the functional role of the identified isoforms in cancer cell lines. We have now expanded and further clarified this important point in the Discussion. In fact, it will be pivotal to check whether the identified isoforms are also expressed in actual tumors to understand their biological significance and possible applications for the therapeutic treatment of cancer. We believe that our work will provide novel insights for the scientific community and prompt further investigation on the role of AS in the regulation of autophagy and its significance in cancer and pathology.

Round 2

Reviewer 2 Report

The authors have addressed to the reviewer requests and improved the manuscript.